# How the Seed of Participatory Plant Breeding Found Its Way in the World through Adaptive Management

**Micaela R. Colley [1,2,*]**, **William F. Tracy [3]**, **Edith T. Lammerts van Bueren [1]**, **Martin Diffley [4]** and **Conny J. M. Almekinders [5]**

1    Plant Breeding Department, Wageningen University, PO Box 386, 6700 AJ Wageningen, The Netherlands; edith.lammertsvanbueren@wur.nl
2    Organic Seed Alliance, Port Townsend, WA 98368, USA
3    Department of Agronomy, Nelson Institute for Environmental Studies, College of Agricultural and Life Sciences, University of Wisconsin-Madison, Madison, WI 53706, USA; wftracy@wisc.edu
4    Organic Farming Works LLC, Farmington, MN 55024, USA; martinvdiffley@gmail.com
5    Department of Social Sciences, Wageningen University, PO Box 8130, 6700 EW Wageningen, The Netherlands; conny.almekinders@wur.nl
*    Correspondence: micaela@seedalliance.org; Tel.: +1-360-531-4043

**Abstract:** Participatory plant breeding (PPB), where farmers and formal breeders collaborate in the breeding process, can be a form of agricultural niche innovation. In PPB, new varieties are commonly adopted by the farmers involved and shared through seed networks, but few are released and commercialized; thus, the variety remains a niche innovation, used within a limited network of beneficiaries. PPB is increasingly emerging to address the needs of organic farmers in the Global North, yet barriers to implementation and institutionalization limit the ability to embed PPB into commercial channels of seed distribution. This case study of a PPB project in the US explores, through the lens of adaptive management, critical points in the commercial release of an organic sweet corn variety, which expanded the innovation beyond the niche environment. The authors show how evolving the actors' roles, expanding the network of participants, and leveraging opportunities that emerged during the process aided in shifting institutional and market norms that commonly restrict the ability to embed PPB varieties in the formal seed system. They further demonstrate that distribution through the formal seed system did not limit access through informal networks; instead, it created a ripple effect of stimulating additional, decentralized breeding, and distribution efforts.

**Keywords:** participatory plant breeding; adaptive management; seed systems; seed networks; niche innovation; organic seed systems; ripple effect

## 1. Introduction

### 1.1. Scope and Relevance of the Study

Participatory plant breeding (PPB) is a type of agricultural innovation, implemented internationally, wherein farmers, professional breeders, and other actors collaborate in variety development, commonly to address the needs of farmers underserved by the dominant seed regime [1–4]. PPB is a promising method to breed for organic systems and address shortages in diversity and quantity of organic seed available [5–8]. Arguments for the methodology include improving heritability by selecting under the environment of intended use, high rates of adoption at lower costs, and developing varieties that can be continually selected to improve adaptation over time [5,6,8,9].

One of the reasons why PPB is a good fit for organic systems is that organic farms are characterized by higher within and between farm agroecological and market variability than conventional farms [5,6,8]. This presents challenges for privately financed breeding programs to recoup expenses in plant breeding and fully serve the scale of the market, as well as the diversity of needs for organically produced seed of suitable varieties [10–12].

While public funding for organic plant breeding increased over the last decade, the short-term nature of public grants often restricts the capacity to deliver finished cultivars to market [7,10–12].

PPB is proposed as a cost-effective model for addressing decentralized organic farmers needs for adaptation to specific agroecological conditions [8,9], yet the barriers to implementation restrict institutionalization, including regulatory constraints, institutional norms, and limitations in financing [7,11,12]. Organic farmers are legally obligated to use organically produced seed, when available in appropriate form, quality, and quantity, as part of their certification requirements in Europe [13] and the US [14]. As such, organic regulations are, in part, stimulating organic breeding activities, including PPB approaches [6,7,11]. The Organic Seed Alliance (OSA) conducts a national survey of organic producers every five years, as part of its State of Organic Seed (SOS) project to track progress, challenges, and opportunities in expanding organic seed use in the United States (US) [11,15,16]. Surveys (conducted in 2010, 2015, and 2020) demonstrated that most organic producers in the US still rely on conventionally bred and produced seed for at least a portion of their crop production. Addressing barriers to PPB presents an opportunity to fill some of these gaps in the organic seed supply.

In most PPB projects, new varieties are commonly adopted by the farmers involved and shared through seed networks or collectives, but often are not formally released and commercialized [7,17]. Projects generally start with agreements on the targeted traits and a breeding strategy but lack a clear strategy for the eventual dispersal and long-term stewardship of the variety, beyond the actors involved; thus, the new variety remains a niche innovation within a limited network of beneficiaries. Commercial seed systems commonly operate within the dominant agricultural innovation paradigm of centralized research and dissemination of innovation, as highly uniform varieties, suited to broad geographical distribution with markets and secured through restrictive intellectual property rights (IPR) [18]. These IPR systems, supported by institutions and laws, implicitly create barriers to development of alternative seed economies. An example is the formal International Union for the Protection of New Varieties of Plants (UPOV) variety registration system, which is enforced in many countries [19,20]. A variety released under UPOV must be highly distinct, uniform, and stable (DUS), traits that are not always valued in organic systems that prize on-farm biodiversity [17,18]. These IPR systems are restrictive for fostering genetic diversity in agroecosystems and often reinforced in the institutional norms of academia, which regulate the IPR of public plant breeders in the US [12,21]. Overcoming these barriers often necessitates a paradigm, where formal release and commercialization are replaced with informal innovation networks, seed is exchanged (rather than sold), and conservation and maintenance are done by farmer innovators.

While innovation networks, based on the exchange of knowledge among diverse actors, is clearly a valuable model for fostering PPB and expanding on-farm biodiversity [18], there is also a need to leverage agroecology and PPB to expand beyond the niche innovation, in order to transform the dominant agro-food regime and expand the diversity of economic models for fostering PPB [22]. Barriers to commercialization by the formal seed registry system in Europe has perhaps incentivized the development of networks of on-farm innovation and exchange of biodiverse seed [17,23,24]. Yet, the recent allowance in the European Union (EU) regulations for organic agriculture to apply for registry of heterogenous varieties may influence the potential for alternative seed economies [25]. Plant breeders in the US are not required to register new varieties in a variety registration system, as the US has not signed the UPOV'91 agreement. This has allowed the emergence of independent farmer plant breeders, commonly releasing open-pollinated varieties through small, regionally-based organic seed companies [26].

### 1.2. Background of the Current Study

In the current study, we explore the experience of an ongoing participatory plant breeding (PPB) project on sweet corn in the US, initiated in 2008 [27,28]. The project

provides the opportunity to analyse the challenges and opportunities encountered in the development, release, and seed distribution of the variety. In particular, the authors identified the critical points in the process of navigating issues of ownership, IPR, financial returns, and variety maintenance, related to the decision to release and commercialize the variety with a seed company, rather than through informal networks. The project members had no pre-determined plan to do so and, in retrospect, were not fully aware of the challenges they would encounter to address institutional barriers and navigate the terms of release to achieve the economy of scale for broad commercial access, while simultaneously fostering access and benefit sharing for smaller regional companies and independent farmer-breeders.

This PPB project was initiated to address organic farmers need for suitable varieties of organic sweet corn seed. Organic acreage of sweet corn in the US is less than 5% of the total seed market, so none of the large vegetable breeding companies are breeding sweet corn nor producing sweet corn seed under certified organic conditions. Typically, in conventional breeding programs, crop nutrition is supplied by synthetic fertilizers; weeds and other pests are controlled with synthetic pesticides, and seeds are treated with synthetic fungicides and insecticides. These options are not allowed under certified organic crop production. This creates two hardships for organic sweet corn farmers: one, they have extremely limited access to varieties that were developed for the very different environmental conditions and management challenges on organic farms; and, two, the inbred parent lines used to produce F1 hybrid varieties often do particularly poorly under organic conditions and, in many cases, are unable to establish a canopy large enough to compete with weeds and produce robust yields. Thus, it is difficult and costly, if not impossible, to produce seed of such varieties under certified organic conditions. When seed can be produced under certified organic conditions, it is often of lower quality and costly, due to the production challenges. The lack of availability of organically produced seed forces organic sweet corn growers to purchase conventionally produced seed that is not treated with seed treatments after harvest.

Organic sweet corn producers are challenged by the lack of access to organically produced seed of appropriate varieties, according to the last three State of Organic Seed Reports. In these national surveys, in the US, organic producers were asked to rank the most important crops and traits to breed for in organic systems. In each year, corn (*Zea mays*) (maize) ranked in the top four priority crops for breeding with yield, flavor, good germination, and disease resistance as top priorities. This is understandable, as well over 95% of sweet corn acreage in the US is devoted to the production of $F_1$ hybrid varieties, and most farmers, shippers, and processors prefer the uniformity offered by these hybrids. Since most of the commercial production of sweet corn uses the $F_1$ hybrid varieties, there is little incentive for seed companies to invest in the development of open-pollinated (OP) varieties. So, except for the small organic breeding community, there is essentially no breeding done for open-pollinated varieties of sweet corn. Many organic farmers are interested in open-pollinated varieties, because they can produce and save their own seed and adapt the variety to their unique conditions. Another advantage of open-pollinated cultivars is that it is often easier to produce seed of OPs than of conventional $F_1$ hybrids under organic conditions. The OP varieties are also more vigorous and compete well with weeds. All of these characteristics are needed for robust seed production under organic conditions. There are a few independent breeders in the US developing open-pollinated sweet corn varieties; however, most of these are in the Pacific Northwest, and the varieties developed (at the time of the current study) were not broadly adapted and well-suited to production in other regions of the country.

The current study is based on a PPB project that was initiated to fill a gap in access to an organically adapted variety of sweet corn, with clear motivations and roles among actors involved in the breeding process [27,28]. Achieving their long-term objectives required strategic decisions to embed the innovation in the broader operating environment, i.e., the seed system. Herein, we identify the critical points in the process of moving from niche

work to a broader operating environment, in order to amplify the scale of impacts, support access to organic seed, and stimulate ongoing breeding efforts. The initial project team, including a formal breeder and associated graduate students, certified organic produce grower, and non-profit project facilitator, evolved over the span of the project, and a broader network of participants developed. The actors developed pathways for the transfer of innovation by applying the concept of adaptive management. Adaptive management refers to the process of creating strategic innovation pathways, based on key decisions at critical points, emerging in the process that encompass social, economic, marketing, psychological, and legal considerations [29–31]. In this case, the effort to release and commercialize the variety raised several issues that the group had not previously considered. Forging a pathway forward required clarifying ownership and IPR, variety commercial release mechanisms, stock seed production, and defining roles after release. The authors elaborate and clarify what these issues entailed and how the group opted for pathways that went in directions different from the current industrial paradigm. In addition, the authors demonstrate how these pathways supported the impact of the project.

## 2. Materials and Methods

In the current study, the authors analyzed the process of the multiplication and diffusion of varieties developed in the ppb project and, thus, covers a time-trajectory after the ppb project itself. The breeding process and methodology of the PPB project is detailed in a prior study by Shelton and Tracy [28]. The current study is recounted on the basis of the experiences of the project facilitator and the breeder (first and second author). The project facilitator, breeder, and associated graduate students kept field notes from meetings and visits conducted during the project. The authors also collaborated with other project members in recounting project details for accuracy and completeness. Project outcomes were further documented by consulting additional actors who entered later in the process of seed production and distribution of the sweet corn variety, including an Australian seed company, an independent farmer breeder, representatives from first company to commercialize the variety (High Mowing Organic Seeds), and the stock seed producer. The seed companies provided sales data to document seed distribution. Media promotion of the variety was assessed through the Meltwater tracking tool [32].

## 3. Results

The results are herein described as three project phases: Section 3.1, actors' motivations in the initiation of the breeding process; Section 3.2, from breeding to commercialization: the choices to be made; and Section 3.3, project outcomes, impacts, and ongoing PPB roles. Four key issues requiring strategic decision-making (choices) are presented in Section 3.2, including: Section 3.2.1, clarifying ownership and IPR; Section 3.2.2, determining the variety release and terms of release; Section 3.2.3, establishing plans for stock seed production and maintenance; and Section 3.2.4, defining the roles of actors, following release.

### 3.1. Actors and Motivations in the Initiation of the Breeding Process

Farmers Martin and Atina Diffley of Organic Farming Works LLC are well-known for their consistent production of high quality organic sweet corn in the Minneapolis, Minnesota region of the US. In 2007, with more than 30 years of experience in growing organic sweet corn at his farm, Martin Diffley shared frustrations regarding sweet corn varieties with breeders Dr. John Navazio (OSA) and Dr. William Tracy, University of Wisconsin (UW)-Madison. Diffley depended on the $F_1$ hybrid variety 'Temptation' for his early season sweet corn production, which was not available as organic seed. 'Temptation', owned by Monsanto (now Bayer), provided superior germination rates in cool spring soil conditions, compared to other varieties he had grown; however, Diffley desired a reliable, certified organic seed source for philosophical and regulatory reasons. Diffley was not alone, as OSA and Tracy heard similar needs expressed by other organic producers. The need was later reflected in the 2010 US State of Organic Seed producer survey, in which

organic producers reported "yield, quality, and emergence" as top breeding priorities for organic corn [15]. There was also a desire to develop sweet corn varieties whose seed could be reliably and cost-effectively produced under organic conditions, in accordance with the United States Department of Agriculture (USDA) National Organic Program (NOP) regulations. OSA and Tracy's program shared the mission of serving the public good by breeding crops with qualities that optimize organic agriculture and expand access to organic seed sources. Thus, Navazio and Tracy proposed (to Diffley) to collaborate on a PPB project to develop an organically bred, on-farm, reproducible, open-pollinated, sugary enhanced (SE) variety with good eating quality, yield, and emergence that Diffley could produce. The actors agreed to collaborate in developing a new variety to suit Diffley's needs, with the shared acknowledgement of the broader goal of expanding access and benefit sharing to additional farmers, in order to maximize positive impacts on organic agriculture. The project launched with the clear goal of breeding a new variety to benefit organic farmers, but with no discussion as to the ownership, name, production, maintenance, or distribution of the new variety.

The partners from the three entities, farming couple from Organic Farming Works LLC, Farmington, MN, USA, and breeders of OSA, Port Townsend, WA, USA, and UW-Madison, Madison, WI, USA, devised a breeding scheme that leveraged their collective knowledge and resources. They collaborated in all phases of breeding and decision-making, i.e., prioritizing traits, making selections in the field, and negotiating the naming and final release of the new variety with shared investments in the breeding efforts. Tracy's program provided the initial germplasm, advised on breeding methods, including utilization of a winter nursery for generating new crosses, and graduate students (Jared Zystro and Adrienne Shelton) to support the breeding activities, including data collection and reporting. OSA's team of researchers facilitated the knowledge exchange and decision-making process and supported the breeding methods, evaluation, and reporting. Diffley led the identification of breeding goals, managed the breeding trials at his farm in Minneapolis, Minnesota, and collaborated in the evaluation of breeding plots. The group looked to Diffley to prioritize traits and assess quality, based on his knowledge of the market standards and agronomic challenges of the farm's environment. The three entities met annually to discuss breeding strategies, evaluate results of the prior year, and set field plans for the following year. The initial material consisted of two breeding populations, each based on intermating four commercial sugary enhancer sweet corn $F_1$ hybrids. One population was roughly five days earlier in maturity and designated 'early', the other designated 'late'. The breeding process followed a recurrent full-sibling selection, with annual evaluation of replicated plots of breeding families on Diffley's farm and regeneration and crossing of remnant seed from selected families at a winter nursery in Chile, managed by Tracy's program. As previously mentioned, details on the breeding methodology and timeline of the first 4 years is described by Shelton and Tracy [28]. The three entities convened each year at peak harvest to evaluate the entire trial, including bite-testing ears from each plot. Disease resistance and agronomic performance were also evaluated [28].

### 3.2. From Breeding to Commercialization: The Choices to Be Made

The three initial entities agreed to share equal decision-making power throughout the release process, as they had from the start, and worked together to devise a strategy that met both the hurdles in releasing a new variety and their collective goals. Unlike many PPB projects, the primary beneficiary, Diffley, had no experience or interest in seed production. This required the actors to consider the impacts of their decisions and adopt new roles to determine the pathway and achieve their objective of broad access to organic seed. The university and non-profit actors needed to fulfill the intent of the funders, USDA Organic Research and Extension Initiative (OREI), and the Organic Farming Research Foundation (OFRF), Santa Cruz, CA, USA, to serve organic stakeholders through broad access and benefit sharing, though only one farmer participated in the initial breeding. The university was also constrained by federal and institutional rules and procedures,

regarding the ownership of IP. Ultimately, the actors recognized the need to engage a broader network of participants to accomplish their goals and address the limits of their collective ability to produce, market, distribute, and maintain seed of the variety. They expanded project boundaries and engaged additional actors to embed the variety within the broader commercial operating environment. The roles of the various project participants, throughout project initiation to variety release, are summarized in Table 1.

**Table 1.** Roles of project participants throughout project initiation to variety release.

| | Farmer (Martin Diffley, Organic Farming Works LLC (Farmington, MN, USA) | NGO (Organic Seed Alliance) (OSA) (Port Townsend, WA, USA) | University (University of Wisconsin-Madison) (Madison, WI, USA) | Seed Company (High Mowing Organic Seed) (HMOS) (Wolcott, VT, USA) |
|---|---|---|---|---|
| Initial project goals | Ensure seed security, crop productivity, and market acceptance. | Expand diversity, quality, and quantity of organic seed for farmers. | Breed for organic and regional needs of farmers. | Fulfill variety needs of organic farmers while supplying 100% certified organic seed. |
| Project roles in participation | Provide field space and farming knowledge. Lead prioritization of traits. Evaluation of breeding lines. | Facilitation of breeding project. Networking with stakeholders outside of project actors. | Provide breeding materials, technology, and infrastructure. | Testing late generation breeding populations with critical knowledge of market demands. |
| Project roles during and after release | Participate in evaluation and field seed selection with stock seed producer. | Negotiate and manage contract terms and financial transactions. Promotion and marketing of variety. Management of stock seed. Continue PPB in sweet corn | Advising on stock seed variety maintenance protocols. Continuing to select and breed divergent populations out of initial project breeding population | Finance seed production and royalties to support ongoing variety maintenance and future PPB projects. Manage variety marketing and distribution. |

### 3.2.1. Clarifying Ownership and IP

Breeding activities were initiated in 2008, with partial funding from OFRF, and continued in 2009, with 4 years of funding from the OREI, as an activity embedded in the broader collaborative project, the Northern Organic Vegetable Improvement Collaborative (NOVIC). This project was subsequently renewed twice, providing 12 years of funding, which allowed the project team to continue breeding new varieties out of the original population for adaptation in diverse regions. The variety released and commercialized through this PPB project was ultimately named 'Who gets kissed?'. At the time of release, the actors did not yet have additional funding secured to support the breeding partners ongoing collaboration in stock seed management and additional participatory plant breeding activities.

All three parties were committed to the concept of open access, which would allow others to use the variety for any purpose they wished, especially adapting the variety to their region and farming system. However, as an employee of the University of Wisconsin, Tracy was constrained by federal, state, and university rules and regulations that made this challenging [6]. It is common in the USA that a contract for a variety release and any royalties are managed not by the breeder but by the university technology transfer department. It is also common in the US university system for administrators to collect a significant portion of the royalties, rather than returning it all to the breeder to support their program. The Wisconsin Alumni Research Foundation (WARF) is the designated technology transfer organization for the University of Wisconsin-Madison, and all potential IP developed at UW-Madison must be disclosed to WARF. WARF has the right of first refusal on all disclosed IP. If WARF refused, then the USDA has the right to choose to claim ownership.

Applying restrictive IP, in the form of a utility patent, plant variety protection, or restrictive license, was not only financially impractical, given the project budget, but antithetical to the project's aim of broad access and benefit sharing and desire to stimulate ongoing breeding efforts, utilizing the heterogenous variety. Upon disclosure, Tracy and WARF officials discussed the unique partnership, variety, and philosophical issues of the organic community, regarding IP. WARF officials chose not to pursue IP on the variety, as did, in turn, the USDA; thus, the rights to release and commercialize the variety was provided to the project partners.

### 3.2.2. Variety Release and Terms of Release

The actors realized the challenges of stabilizing an open-pollinated population and recognized that achieving the uniformity level of an $F_1$ hybrid was not possible. Thus, the team needed to determine when the population was uniform enough and of high enough quality to release as an open-pollinated variety. Release of a heterogenous variety does not present a regulatory problem in the US, as there is no formal registry system, as in Europe and elsewhere. The group, with Diffley's lead, decided that, when 75% of the ears were of exceptional size and eating quality to meet Diffley's premium market, then the population was ready for commercial production and release. In the 2013 evaluation of the breeding populations, the partners collectively determined that the late maturing breeding population had reached this point and it was, thus, ready for release [28].

The network of collaborators involved in the NOVIC project served as a testing network, with on-farm and on-station variety trials in four states across the Northern tier of the US (Oregon, Washington, Wisconsin, and New York). This provided the opportunity to assess performance across other Northern environments and raise awareness of the new variety. The NOVIC network exchanged varieties and shared trial results with organic seed companies, in addition to farmers. The breeders provided samples of the sweet corn to companies for trial, and one company, in particular, expressed interest in commercialization, High Mowing Organic Seeds (HMOS), located in Hardwick, VT, USA. This company only sells certified organic seed and distributes nationally. At that time, there was not organically bred $F_1$ hybrid sweet corn varieties on the market, and the open-pollinated varieties were highly variable and lacked consistent yield and quality. Thus, a new open-pollinated variety, of commercial quality that they could produce and sell would fill a gap in their market. There were no regional seed companies or farmers who expressed interest in producing the seed at that time. The actors agreed to explore partnering with HMOS to commercialize the variety but needed to carefully consider the terms of a contract for commercialization, in order to ensure they could achieve the project goals of wide accessibility, as well as the breeders' ability to remain involved in variety maintenance through this pathway.

The actors had to consider that the public funding source supported the breeding costs, but it did not support the costs of commercialization or ongoing stock seed management. The actors also did not know if there would be another grant cycle to support ongoing breeding with the two populations. Recognizing their dependance on a single, unstable funding source motivated the actors to explore the potential to recoup a financial return on commercialization, in order to support their ongoing PPB efforts.

The three partners agreed to equally share in any revenues. The actors negotiated for royalties on commercial sales, without IPR to support their ongoing involvement. For the seed company to ensure enough sales to support their investments in marketing and production they requested exclusive access to the stock seed and asked that it not to be released to other companies, at least for the first three years. This agreement provided a sales advantage to the seed company and allowed the breeders to work directly with one company in the management of stock seed. The breeders were concerned that selling to only one company would too narrowly restrict access to the diversity of scales of organic seed companies emerging and serving regional markets. Thus, the company agreed to sell wholesale quantities of seed of the variety to smaller companies for repackaging to extend the channels of distribution. Without IP, the variety could also clearly be purchased for the purpose of regenerating for on farm use and/or for additional breeding efforts.

### 3.2.3. Seed Stock Production and Maintenance

The quality and stability of varieties of many crop species may be managed with minimal selection toward the ideotype, commonly managed by the seed company. This is true of highly self-pollinated crops, such as common bean, tomato, and oats. Cross-pollinated crops, such as sweet corn, spinach, beets, and the cucurbits, are heterogenous and heterozygous and require considerable diligence to guard against outcrossing with foreign pollen, inbreeding, and natural selection away from quality traits. Given the demand in the

organic community for non-GMO crops, as well as the free crossing nature between sweet corn and grain corn pollen, contamination is also of great concern. The group realized that increasing and maintaining the new variety would require continual monitoring and selection to ensure no genetic drift or contamination occurred. This factor weighed on the actors' negotiations of terms of release and commitments to continuing collaborating in the ongoing stock seed management. The actors desired to work directly with a stock seed producer, with secure isolation from GMO corn crops, who also held an interest in collaborating in the monitoring and selection of the variety. They identified an ideal sweet corn seed producer and negotiated with HMOS to contract for seed production with this farmer. The three actors committed to visiting the farmer during production and directing the seed selection process. They also committed to routinely screening and selecting the stock seed, in a high disease pressure environment (Madison, WI, USA), every few years, to ensure the seed was produced under low disease for quality purposes (Dixon, MT, USA) but resistance maintained by periodic selection under the high-pressure environment. Commitment to these activities contributed to the need for additional funding the support their costs of involvement.

### 3.2.4. Defining Roles following Release

The parties drafted a contract, stipulating the terms of release, that addressed exclusivity, royalties, wholesale, and retail sales. It also specified collaboration between the company and breeders in stock seed management, promotion, and marketing. The contract clarified that the co-breeders served as equal parties (OSA, UW-Madison, and Organic Farming Works LLC), and they agreed to equally share in the royalties to support their continued collaboration. The non-profit served as the fiscal entity, for the purpose handling the contract and associated costs of stock seed production and distribution of royalties. In this instance, the non-profit's freedom to operate facilitated the unconventional participatory process and shared benefits.

The breeders and seed company also shared concerns of market acceptance, as the ears were more variable than a $F_1$ hybrid, with mainly white and yellow bicolor kernels and occasional light pink ones. Thus, they carefully considered the naming, storytelling, and promotion of the variety, to acknowledge that a variety could retain a level of diversity, while providing a quality product. The challenge of public acceptance of intravarietal diversity presented the opportunity to educate on the value of genetic diversity and long history of farmers' role in improving, adapting, and stewarding populations from land races to heirloom varieties. The non-profit, university breeder, and seed company worked together in their promotion of the project to recount the farmer-centric participatory process and value of diversity, adaptability, and farmer-stewardship in organic systems. They also chose the name, 'Who gets kissed?' to reflect the historical acceptance of diversity in a variety. Historically, many communities collaborated in annual seed harvests and sometimes played games to keep the work fun. One big community task was to remove the husks on the ears of corn after harvest, so they could be stored for winter. The story goes that one playful version of the husking circle was that whoever husked an ear that had a red kernel amidst the white and yellow rows got to pick who to kiss in the circle. This lighthearted game reflects the historical acceptance and even celebration in retaining genetic and phenotypic diversity within a variety. The actors collaborated in developing a press release, social media, and marketing materials that promoted the participatory breeding process in variety improvement and retention of biodiversity. The media picked up the story, resulting in more than 100 media articles reporting on the project's story across the USA [31].

### 3.3. Project Outcomes, Impacts, and Ongoing PPB Roles

In the first year of sales, 'Who gets kissed?' brought in the highest recorded sales of a new release at High Mowing Organic Seeds within the first year than any previous new release from the company. Sales went to 49 US states and Canada, with more than 2500 kg

sold by 2020. Wholesale distribution resulted in sales by at least 14 regional seed companies in the US. The breeding team continued to maintain the variety quality through trials and stock seed production, in partnership with an organic seed grower in Montana, USA. In 2018, an organic seed company, by the name of The Biodynamic Seed Company, in Australia, trialed 'Who gets kissed?', as they were seeking an open-pollinated sweet corn to produce for the Australian organic seed market. In Australia, all imported corn seed is required to be treated with chemical fungicides prior to import, and there are no domestic companies breeding or producing sweet corn seed for the organic market. Thus, an open-pollinated variety of high quality was desired for domestic production. 'Who gets kissed?' performed very well in their trials, and the company requested approval to produce and distribute the variety to wholesale and retail seed companies. The company willingly offered 10% royalties, recognizing the value of supporting the ongoing participatory breeding efforts. Production and commercial sales launched in Australia in 2021, with 21 kg sold in the first year.

After the release of 'Who gets kissed?', the Organic Seed Alliance and UW Madison breeders continued to collaborate in farmer-participatory sweet corn breeding for organic production. The initial breeding process resulted in two distinct, but related, populations, differentiated by the timing of maturity (early versus late). As previously mentioned, 'Who gets kissed?' was derived from the later-maturing population, which suited the climate of the Upper Midwest region, where Diffley farmed, as well as many other regions of the US. However, it was too late maturing to suit producers in mild climates, such as the northern maritime Olympic Peninsula of WA (US), where the Organic Seed Alliance collaborated with farmers in on-farm breeding. Farmers in this region were similarly dependent on 'Temptation', which concerned the local organic foods cooperative that purchased produce from local growers. Thus, the Food Co-op of Port Townsend, WA, US, provided an initial year of funding for the team to launch a participatory breeding project, which they called "Olympic Sweet", utilizing the "early population", and later continued under the scope of the NOVIC project. Three local farms collaborated in the Olympic Sweet breeding project, including an educational farm that integrated the half-sibling selection methods into their applied farmer training program. In Wisconsin, Tracy continued selecting out of 'Who gets kissed?', in order to shift the population to achieve an earlier and more uniform timing of maturity. The variety is tentatively called 'Who gets kissed too?'.

Tracy has also developed a variety from the early population, called 'Quick Kiss'. Additionally, at least three farmer-breeder projects utilized 'Who gets kissed?' and the early population as a breeding parent, and one was released in 2020 as a new variety, 'Sweet kisses', pledged under the Open Source Seed Initiative [33]. Actors' motivations, from a philosophical, economic, agronomic, and practical point of view, influencing their decisions for the variety release pathway, are summarized in Table 2.

**Table 2.** Actors' motivations, influencing decisions for variety release pathway.

|  | **Philosophical** | **Economic** | **Agronomic** | **Practical** |
|---|---|---|---|---|
| Farmer, Organic Farming Works LLC | Preference for organic seed and avoiding seed from companies that sell GMOs. | Compensation necessary to continue aiding in variety selection to maintain quality. | Seed available in adequate quantity, quality, and price to serve needs of commercial-scale organic producers. | Needs someone else to manage seed production and distribution, but willing to continue participating in stock seed selection. |
| Plant breeders of UW-Madison | Serve public good by making the variety accessible to farmers and breeders for ongoing variety improvement efforts. | Financial returns needed to support ongoing breeding and variety maintenance and improvement work. | Important to maintain qualities of good emergence and eating qualities. | Can provide field space for variety evaluation and continued breeding, but not for seed production. |
| NGO (OSA) | Avoid restrictive IP. Expand access to organic seed of the variety to maximize impacts. | Financial returns necessary to support NGO involvement in education on the variety and breeding process and support of stock seed maintenance. | An organically available sweet corn reduces farmers' dependance on non-organic seed sources. | Able to advise seed producer and seed company and support promotion of variety through press releases and other media. |

## 4. Discussion

The current study demonstrates how the adaptive management a PPB product can develop from an agricultural niche novelty into an innovation that is embedded in the broader environment to achieve economies of scale and support the sustainability of a PPB program. The authors show that addressing the emerging obstacles, responding to opportunities, and expanding the roles of actors and the network of participants was necessary in shifting institutional and market norms that commonly restrict the ability to embed PPB varieties in the formal seed system. This required negotiating with external policy makers, such as university administrators, and creatively developing alternative pathways for seed production and distribution. The actors also had to consider their own roles, beyond the breeding process, and commitments to the project long term. The variety release pathway in the current study was forged in response to the unique context and circumstances the actors faced. While the experience was unique, the lessons learned reinforce that PPB is, by nature, a dynamic innovation process, based on knowledge exchange and activities, tailored to address the context specific needs and abilities of participants. For this reason, being responsive, creative, flexible, and willing to engage with unusual commercial partners, i.e., being adaptive in the management of the process, is critical in innovation systems, such as PPB, that do not follow a prescribed pathway for the transfer of innovations. 'Who gets kissed?' achieved commercial success and sustained sales over the past 8 years, but that is only one lens of success.

The non-restrictive release was critical to ensure that distribution through the formal seed system could be managed in a manner that did not limit access through informal networks, but instead created a ripple effect of stimulating additional, participatory, and independent breeding and distribution efforts. While the current PPB project served the participating farmer's needs and led to breeding additional varieties for diverse climates, the authors acknowledge that the agronomic value of the variety is limited in scope. There are more challenges to be faced when varieties spread to other parts of the world, where growing conditions, pests, diseases, and consumer preferences differ from those in the target region. The involvement of only one farmer in the initial breeding project presented a risk of limiting project beneficiaries and, as pointed out by Chiffoleau and Desclaux [5], potentially strengthening the power of decision-making in select farmers of high socio-economic status, who are not representative of all stakeholders concerned. This consideration weighed upon the actors' motivations to ensure the variety was released in a manner that expanded access and stimulated additional farmer-participatory and independent breeding efforts.

The concept of ownership and IP, in this case, as with most PPB projects, is contrary to the intent of the PPB program serving the public good. As such, the project partners chose to develop a formula that combined broad access with benefit sharing. The reality is that many public breeding institutions are under-funded and encouraged by policy makers to pursue private investments and royalties on innovations to support not only their programs but institutional administration, as well. In this case, the formal breeder was able to convince the university officials that IP was not appropriate. The participatory nature, with multiple actors undertaking the breeding activities, and program goals of the government funding agency helped influence the university's decision not to pursue IP. In this case, the university breeder also held seniority in the department, and the authors acknowledge that the ability to negotiate is not always equitable in university systems. The authors have heard similar reports from other university breeders that if the farmer is a primary decision-maker in on-farm breeding, rather than the breeder, then determining ownership is much more ambiguous, resulting in university officials opting not to pursue IP. This underscores the need to address university policies, if we are to institutionalize PPB in public programs.

The opportunity and decision to collaborate with a willing commercial partner, who committed to marketing the seed and was willing to share benefits, was an important feature in the sustainability of this PPB initiative. Many PPB programs depend on grant

funding, as in the current case, and limits in grant timelines often limits the ability to see projects to full fruition [7,11,12]. Fortunately, in the case of NOVIC and the PPB sweet corn project, the funder (OREI) allows projects to apply for up to three renewals, recognizing the long-term nature of plant breeding. At the time of the release of 'Who gets kissed?', the actors did not know if renewed funding would be granted, influencing the desire to ensure some economic return to sustain their participation. Ultimately, the royalties alone would not have sustained the extent of the continued breeding work, but it did provide financial flexibility to support stock seed production and marketing efforts. Ironically, royalty funds also covered the costs of open access publishing of the current article. In the current study, the willingness of the seed company to provide royalties, without restrictive IP, is a promising precedent in supporting PPB. There is evidence that this arrangement is becoming more common in the US, with organic and regional seed companies supporting independent breeders through royalties, as well [26].

Similar to many PPB programs, one of the primary goals of the initiative was the development of genetically diverse, open-pollinated populations that could be selected to evolve over time and adapted to new environments. The actors contended with the challenge, encountered by many other PPB practitioners, of defining the ideotype of a variety and assessing marketability, when the goal of uniformity is moderated by breeding objectives of adaptability, resilience, and retention of biodiversity [5,34]. In this case, the commitment of the seed company to promote the virtues of PPB and value of diversity in their marketing enabled the commercial success. This demonstrates how the engagement of diverse actors in a value chain can be instrumental to promote novel and heterogenous varieties and support the ability to embed PPB in the market economy. The importance of actors that can successfully access the market has also been crucial in other PPB cases, albeit in different forms. In the PPB case of the late blight-resistant potato in the Netherlands, the seed companies were already involved in the PPB program as actors; however, the missing actors to adopt the new PPB varieties were supermarkets. A separate action was needed by the Dutch umbrella organization for organic agriculture to convince retailers of the value of the new disease-resistant PPB varieties, which ultimately succeeded in creating a covenant among all Dutch supermarkets to replace the current late blight susceptible varieties with more robust PPB varieties over time [35]. In other PPB projects, aiming at resistant varieties, such as scab-resistant apples and mildew-resistant grapes, benefitting farmers in the first place, demonstrated the need to find alternative ways to involve the market actors in accepting the new varieties and communicate the added value to consumers [36].

If the end goal of a PPB program is to expand farmers' access to improved varieties, then projects must be embedded within a seed system, whether formal or informal. In some cases, the seed system may be maintained through on-farm seed saving or managed through a seed network [7,23,24]. This case, and others, show that commercialization may be accomplished by a variety of business models. In any case, if the distribution and/or commercialization pathway is not considered or developed over time, then the improved variety is at risk of limited adoption and eventual loss. PPB is a dynamic process that encompasses a diversity of models, operating within varied socioeconomic contexts [5,7,37]. Thus, the pathway to embedding PPB innovations in the seed system must address the unique context and circumstances. As the current study demonstrates, reflexively adapting as the pathway unfolds and partners face the emerging challenges can be a successful way of addressing the contextual uniqueness [29,30]. In this case, the commercial pathways and informal networks were able to operate complimentarily, in order to benefit stakeholders of varied scales, simultaneously supporting production economies and expansion of agrobiodiversity. Similarly, Li et al. [37] described how public and private interests may work together to simultaneously breed $F_1$ hybrid varieties, as well as conserve and improve farmers' land race varieties through PPB. The current study reinforces prior experiences by contributing an additional example of how navigating the institutional and policy constraints can be overcome to integrate PPB varieties into commercial environments, without excluding the potential for supporting farmer-centric

PPB and independent breeders in the process. The authors hope the experience and lessons learned inspire other PPB practitioners to apply adaptive management concepts, within their own contexts, to navigate obstacles and respond to opportunities necessary, in order to realize the potential of PPB as an innovation pathway for agroecological seed systems.

**Author Contributions:** The authors contributed to the research and authorship in the following roles: conceptualization, M.R.C., W.F.T., E.T.L.v.B., M.D. and C.J.M.A.; methodology, M.R.C., W.F.T. and M.D.; formal analysis, M.R.C., W.F.T., E.T.L.v.B. and C.J.M.A.; resources, M.R.C., W.F.T. and M.D.; writing, review, and editing, M.R.C., W.F.T., E.T.L.v.B. and C.J.M.A.; supervision, W.F.T., E.T.L.v.B. and C.J.M.A.; project administration and funding acquisition, M.R.C. and W.F.T. All authors have read and agreed to the published version of the manuscript.

**Funding:** This research was funded by the USDA/NIFA/OREI, grant numbers: 2009-51300-05585 for NOVIC 1, 2014-51300-22223 for NOVIC 2, and 2018-51300-28430 for NOVIC 3, as well as the Organic Farming Research Foundation, and the Food Coop of Port Townsend, WA, USA.

**Institutional Review Board Statement:** Not applicable.

**Informed Consent Statement:** Not applicable.

**Data Availability Statement:** Not applicable.

**Acknowledgments:** The authors gratefully acknowledge the hard work, critical thinking, and research support of graduate students Adriene Shelton, Jared Zystro, and Tessa Peters. The authors acknowledge the inspiring initiative of John Navazio and Jared Zystro for helping launch the project and securing initial funding. The authors thank James Myers for his 12 years of leadership of the NOVIC project and all NOVIC partners. Lastly, the authors wish to express their gratitude to High Mowing Organic Seeds and the Port Townsend Food Coop for the financial and philosophical support of participatory plant breeding and organic seed systems.

**Conflicts of Interest:** The authors declare no conflict of interest.

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
