# Peer review of "How the Seed of Participatory Plant Breeding Found Its Way in the World through Adaptive Management"

_sustainability, doi:10.3390/su14042132_

Round 1

Reviewer 1 Report

Please consider the following minor revisions.

Ln 71 "Inernational" to correct by "International"

Ln 80 "are doen by" to correct by "are done by"

Ln 427 “deveoped” by “developed”

Ln 477 “sweete” please see if you mean this or “Sweet”

Author Response

Response to Reviewer 1 Comments

Point 1: Please consider the following minor revisions:

Ln 71 "Inernational" to correct by "International"

Ln 80 "are doen by" to correct by "are done by"

Ln 427 “deveoped” by “developed”

Ln 477 “sweete” please see if you mean this or “Sweet”

Response 1: All of the edit suggestions are typos and are corrected as “International”, “are done by”, “developed”, and “sweet”.

Reviewer 2 Report

The submitted manuscript is based “on a PPB project that was initiated to fill a gap in access to an organically adapted variety of sweet corn with clear motivations and roles among the actors involved in the farming process”. In the paper, “the authors seek to identify the critical points in the process of moving from niche work to a broader operating environment to amplify the scale of impacts, support access to organic seeds and stimulate ongoing breeding efforts.” However, the work requires some revisions: 1. The PPB project on which the study is based should be presented and detailed in the paper 2. The results should include qualitative / quantitative data 3. The results should be discussed in more detail

Author Response

Response to Reviewer 2 Comments

Point 1:  The PPB project on which the study is based should be presented and detailed in the paper

 Response 1: In the current study, the authors analyzed the process of the multiplication and diffusion of the varieties developed in the ppb project and thus covers a time-trajectory after the ppb project itself. The authors have added clarity on the scope of the current study in the materials and methods section. Background on the project is provided in the introduction and detailed in the results section. The breeding process and methods are cited as a previous study (Shelton and Tracy, 2015) (Line 235).

 Point 2:  The results should include qualitative / quantitative data

Response 2: The results describe the outcomes of the application of adaptive management in navigating the commercialization process in a qualitative way. There are few aspects in this process that suit a quantitative description other than the impacts of the project as we have added.

Point 3:  The results should be discussed in more detail

Response 3: The authors have added additional details in the discussion.  

Point 4: “Is the article adequately referenced?” reviewer response - “can be improved”.

Response: Additional referencese are added throughout the paper.

Reviewer 3 Report

Dear authors, congratulations for your work and research!

Author Response

Response to Reviewer 3 Comments

Point 1:  Congratulations for your work and research!

Response 1: Thank you. We hope you and others find it valuable in development of future PPB programs.

Reviewer 4 Report

The study is well-conceived and the idea where farmers and breeders collaborate in the breeding process is interesting for the future of organic agriculture. However in my opinion there will be challenges concerning plants resistance at stem borer attacks. For example, sweet corn is more sensitive to ECB (European corn borer) attacks. 

Author Response

Response to Reviewer 4 Comments

Point 1:  The study is well-conceived and the idea where farmers and breeders collaborate in the breeding process is interesting for the future of organic agriculture. However in my opinion there will be challenges concerning plants resistance at stem borer attacks. For example, sweet corn is more sensitive to ECB (European corn borer) attacks. 

Response 1: Thank you for your review. Indeed there is a wide range of methods and roles that formal researchers and farmers apply in various PPB programs. Sweet corn differs in many ways from field corn including pest resistance. This is true regardless of the breeding method. In the USA many organic farmers grow sweet corn and used organic methods to minimize damage. In the case of sweet corn in this region of the US stem borer was not a primary concern of the participating farmer.  There are undoubtedly more challenges to be faced when this variety is spread to other parts of the world where growing conditions, pest and diseases and consumer preferences differ from those in Northern America. We have elaborated on that fact in the discussion.